# Origami folding: Taxing resources necessary for the acquisition of sequential skills

**Fang Zhao**[1]*, **Robert Gaschler**[1], **Anneli Kneschke**[1], **Simon Radler**[1], **Melanie Gausmann**[1], **Christina Duttine**[1], **Hilde Haider**[2]

**1** Department of Psychology, University of Hagen, Hagen, Germany, **2** Department of Psychology, University of Cologne, Cologne, Germany

* fang.zhao@fernuni-hagen.de

**Data Availability Statement:** All experiment and data files are available from the Open Science Framework database (https://osf.io/p3tyf/).

## Abstract

Sequential skill learning with practice is fundamental to human activity (e.g., tying shoes). Given the lack of prior knowledge in most participants, Origami folding is a promising task to study the acquisition of a sequential skill. While previous Origami folding studies mainly dealt with the question, which forms of instruction can lead to better learning outcomes, we employ a dual-task approach to test which resources are necessary for folding and for improvement with practice. Participants ($N = 53$) folded five Origami figures for four times each, which were randomly paired with five types of secondary tasks to cause either cognitive (verbal vs. visuospatial) or motoric (isochronous vs. non-isochronous tapping) memory load or none (control condition). Origami performance showed a typical learning curve from Trial 1 (first run of folding the figure) to Trial 4 (fourth run of folding the same figure). We tested for a dissociation between variants of memory load influencing Origami folding performance vs. the variants influencing learning (i.e. change in performance across practice). In line with theories suggesting that learning operates on the level that (at a given point in practice) demands the most control, we did not observe cases where a dual-task variant influenced performance while it did not affect learning. Memory load from the cognitive visuospatial secondary task as well as the isochronous tapping secondary task interfered with improvement in Origami folding with practice. This might be due to the use of visuospatial sketchpad and absolute timing mechanism during the acquisition of Origami folding.

## Introduction

Many of our everyday life skills are determined by a certain sequence [cf. 1]. Origami paper folding also consists of performing series of actions following a specific sequence. If one step is wrong or ignored, it is very unlikely that the figure can be folded correctly. Studies in movement science show that the motor and cognitive control in sequential skills often interfere with each other in a trade-off manner. For instance, attentional resources are allocated with prioritization to the postural task to avoid falling in elderly people [2]. There is a degraded performance in postural control when participants are concurrently solving a spatial working memory task [3].

**Funding:** The author(s) received no specific funding for this work.

**Competing interests:** The authors have declared that no competing interests exist.

Under dual-tasking, selective memory load can affect performance due to our limited processing capacity in working memory [e.g., 4–6]. Yet the cognitive resources needed to perform the sequential skills might be different from the cognitive resources required to learn the sequential skills, which can be measured by the improved performance across trials with practice. Past studies suggest that motor learning can work even better when there is concurrent cognitive load. Langhanns and Müller [7] documented better performance with paddleball task (fast rhythmic movement) under dual-tasking compared to the single-task condition. Better performance in circle drawing was observed in the dual-tasking condition compared to single-tasking [8]. Decreased time variability of finger tapping was reported under dual-tasking compared to single-tasking [9]. Blanchard et al. [10] showed an increase in body sway (indicating degraded performance) when standing still compared to standing still while concurrently performing a secondary task (reading aloud or counting backward). Presumably, cognitive monitoring is harmful for some motor control process [cf. 11, 12]. Some motoric sequences rely on fine-grained movement patterns involving smooth and temporally accurate processing. Cognitive monitoring can interfere with motor processes that rely on subroutines acquired earlier. Therefore, relying on cognitive monitoring for more than initial scaffolding of the configuration of motor control might delay shifting control to the adequate level. Adequate secondary memory load might reduce the capability to engage in cognitive monitoring and drive participants to rely on automated subroutines more quickly. In this study, we examined whether and which type of memory load interferes with the performance and learning (change in performance across practice trials) of a sequential skill.

As cognitive control and motor control are required to fold Origami [cf. 13], a secondary task is added so that cognitive or motor capacity can be loaded. Participants folded each figure four times. We analyzed the change in performance across the four trials in folding the same Origami in order to examine which resources are necessary on performance vs. acquisition of sequential skills. How well one acquires the skill of folding can be estimated by change in performance across practice trials. If a specific secondary task disrupts Origami folding, it will be assumed that folding and the secondary task use overlapping resources. Potentially, resources required for improving Origami folding from one trial to the next do not completely overlap with the resources required for executing the sequential skill. This would show by differences in which secondary tasks have an impact on skill performance vs. skill acquisition (changes in performance through practice).

## Origami folding task

So far the effects of memory load on sequential skills have been tested by using tasks like walking [2], paddleball or pegboard [7], or ankle movements [14]. Our decision to combine an Origami folding task with various memory load tasks was guided by three motives. First, despite that Origami folding has been spread out through the world for a long time, many people are novices (different from e.g. car driving) so that acquisition of a sequential skill can be studied. The long history might also help to motivate novices to learn the skill. It is believed that paper folding developed after the invention of paper in 105 A.D. in China and was spread by Buddhist monks through Korea to Japan in 6[th] century [15]. Nowadays most people have easy access to paper and most have experiences in folding planes or cranes. Second, Origami folding consists of common features of sequential skills and has been used to study the acquisition of sequential skills [16–18]. People execute hierarchical action plans, which decompose an overreaching goal into sub-goals, to organize behavior [cf. 19]. Third, unlike driving or playing piano, paper folding experiments require only papers and are ideal for research projects with tight budgets.

Tenbrink and Taylor [13] proposed four stages in Origami paper folding by analyzing the verbal protocols during the folding task. The first stage is *reading*. At this stage, participants read the instruction, such as an image showing "fold two corners under the inside part of the wings". Some people start directly with the second stage *reformulating*. They interpret the meaning of the instruction in their own words. According to the results of Tenbrink and Taylor [13], the third stage is *reconceptualizing* by adding ideas about the folding step. During reconceptualization, participants use many spatial concepts "in, on, to, up, middle". The authors furthermore discuss the relevance of processing crease, alignment and orientation of the objects. This highlights the mental operations necessary to move from two-dimensional instruction to three-dimensional product. The last stage is the *evaluation* of the product in comparison to the instruction.

The underlying mechanism of learning Origami is motor skill learning, marked by increasingly accurate movements in space and timing with practice [12]. It is a fundamental ability of human activity. Without motor skill learning, even simple tasks like tying shoes can cost a large amount of attention each time. In accordance with Willingham [12], two mechanisms facilitate motor skill learning. On the one hand, each time a task is executed, the processes of motor control (i.e., perceptual-motor integration, sequencing and dynamic processes) can be tuned and transformed more efficiently. On the other hand, the strategic process can support motor skill learning by selecting goals that are more effective or selecting and sequencing targets that are more effective for movements. In addition, the mechanism involved in learning the sequential skill are related to the Mirror Neuron System [20, 21]. It is a system that binds motor perception and motor practice, which can be activated by imagining an action or observing a human movement.

The current study employed paper folding to test how sequential skills are acquired with practice. Previous Origami folding studies mainly dealt with the question of which forms of instruction can lead to better learning outcomes, e.g., animation vs. static graphs or text only vs. text with graphs [16–18, 22, 23]. However, the studies have not addressed the question *how* sequential skills are acquired under repetitive practices (i.e., time course of practice-related improvements, impact of memory load as an indicator of potential resources involved). Since we all have the experience to learn sequential skills (e.g., dancing, cooking) with more than one attempt, it is important to investigate which type of resources are involved in the acquisition of sequential skills with practice.

## Dual-task interference

Atkinson and Shiffrin [24] suggested that the human cognitive architecture includes a working memory with limited capacity and an extensive long-term memory. Recent works suggest analogous mechanisms of selection and updating in declarative and procedural working memory [25]. Working memory capacity has been estimated to be four items [e.g., 26] and has been discussed with respect to implications in learning settings [see a review: 27]. One influential model suggests that working memory consists of a central executive, a phonological loop to manipulate the verbal information, and a visuospatial sketchpad to store imagery [28]. The visuospatial sketchpad might be relevant in Origami folding as participants are asked to comprehend the pictures of folding steps. Although learning Origami folding is a pictorial task, it can still interfere with the phonological loop. Previous studies on implicit sequence learning [29, 30] have shown that an auditory (auditory-verbal) task can significantly interfere with the acquisition of a sequential skill in a spatial (visual manual–screen locations to key locations) task. The cognitive processes involved in Origami folding [13] include that participants reformulate and reconceptualize the instruction in their own words or thoughts to conduct the

movements. During the comprehension of pictures, the perceived information from images in pictorial channel can be converted to the phonological loop in the verbal channel [cf. Integrative Model of Text-Picture Integration, 31]. When the mental representation of the instruction is constructed, the information can be converted to verbal as well as pictorial information. For example, after seeing the picture of one folding step some participants might use their own words to explain which action they should perform. Origami folding would therefore make use of the visuospatial sketchpad as well as the phonological loop and the central executive to transfer information between the two storages. A secondary task could demand resources that are lacking for the information transformation processes necessary for folding and necessary to assign credit to particular steps in the folding process in order to learn to improve performance [cf. 32].

Sequential skills often involve a trade-off between cognitive and motor control [see a review, 33]. The *Motor Simulation Theory* [34, see a review, 35] proposes that the human motor system is part of a simulation network. It consists of motor imagery (mental representation of the action without engaging in the movement) as well as motor execution. The motor imagery rehearses motor-related information off-line by simulating the action execution and finally configures the motor system according to the intended actions. Motor imagery and action execution hence can share common mental representations and mechanisms. Take Origami folding as an example, participants use motor imagery to animate the static folding steps without performing any movement. Then they execute the action by translating the motor imagery into folding actions. A secondary motor task could request resources that are necessary for the motor imagery and action execution of Origami folding. Thus, secondary task load might influence performance and learning. Furthermore, learning might ease the impact of a secondary task. To the extent the sequential skill is learned (i.e., information is stored in motor chunks [cf. 36, 37] or is stored as higher order units called cognitive schemata, [cf. 38, 39]), people might need to rely less on instructions and convertibility between verbal vs. visual format or cognitive vs. motor system. This would leave more resources for the secondary task.

When we perform multiple tasks simultaneously, the execution of the tasks normally slows down due to dual-tasking interference. There are many hypotheses explaining this issue [see a review: 40]. The single channel *Bottleneck hypothesis* [41–44] presumes that information processing has a bottleneck or filter with limited capacity. The *Attentional Resource theory* [45] suggests that the limited capacity central processor can, to some extent, be shared between tasks. The mental resources can be allocated to one task and the leftover resources can be applied to the other task. However, the evidence of material-specific trade-offs between the primary and the secondary tasks leads to a new perspective: The *Multiple Resource theory* [46–48]. It assumes that two parallel tasks only interfere with each other when using the same mental resources. Due to the evidence of verbal-pictorial dual-channel process [e.g., 28] and the trade-offs between cognitive and motor control [e.g., 33], the Multiple Resource theory is used to explain how memory load affects the acquisition of sequential skills.

## Cognitive and motor control

According to the Baddeley-model [49], dual task paradigms can lead to resource conflicts in the phonological loop or in the visuospatial sketchpad. By testing participants on a task paired with either load for the visuospatial sketchpad or load for the phonological loop, researchers can check which representation format their task of interest seems to be mostly drawing upon. For instance, when asking participants to memorize chess configurations, impairment was revealed for the visuospatial sketchpad only [4]. Similarly, only interference with a visuospatial working memory load task was revealed when participants performed a mental animation task

and a visuospatial vs. verbal memory load task concurrently [6, cf. 50, 51]. We thus used cognitive *verbal* and cognitive *visuospatial* secondary tasks [52] to test which representation format was mostly relevant for Origami folding and improvement in folding.

According to the Multiple Resource theory [48], dual-task paradigms can lead to resource conflicts between cognitive and motor control. Finger tapping of interval patterns is used in the experiment to examine the motor control under dual-tasking [e.g., 53, 54]. When perceiving the interval patterns, people attempt to generate an internal clock, which enables the accurate representation of temporal structure [55]. Whether the internal clock is successfully used or not depends on the structure of the rhythmic patterns, which have been differentiated into patterns with isochronous structure (inter-stimulus onset intervals, IOIs, of equal duration) or non-isochronous structure (IOIs of arbitrary durations) [56]. When the rhythmic patterns are isochronous, the clock can be evoked and the temporal structure can be represented accurately. In contrast, when the clock is not evoked, the temporal structure is not represented accurately and reproduction is based on the assimilation and distinction principles [57–60]. The assimilation principle refers to the tendency to equalize similar temporal intervals. For example, people tend to assimilate rhythmic patterns with 4:5 ratio (e.g., 400ms vs. 500ms) towards a 1:1 ratio. The distinction principle refers to the tendency to distinguish different temporal intervals by categorizing them into long vs. short intervals [61–64]. For example, rhythmic patterns with 3:2 ratio (e.g., 900ms vs. 600ms) tend to be reproduced with a long/short ratio of 2:1 by exaggerating the longer intervals to duple multiple of the basic interval. Taken together, the absolute timing mechanism (i.e., exact temporal values) is prompted with isochronous structure. The relative timing mechanism (i.e., relative temporal ratios) is evoked with non-isochronous structure. As sequential skills involve movement in accurate timing [12], we speculated that tapping rhythmic patterns with isochronous vs. non-isochronous patterns would to a different extent overload the limited resources in motor modality, leading to decrements in improvements of Origami folding.

### Research question

In this article we aimed at comparing how different variants of memory load affect performance in Origami folding and acquisition of Origami folding (change in performance across practice trials on a specific figure). To this end, we compared the course of the Origami performance from Trial 1 to Trial 4 accompanying four variants of memory load (cognitive verbal, cognitive visuospatial, motoric isochronous tapping, motoric non-isochronous tapping) with single-task Origami folding.

The abovementioned studies suggest that different perspectives on the resources necessary for Origami folding need to be tested by using different variants of dual-task load. In particular, we hypothesized that visuospatial load and timing load should interfere with practice-related improvement in Origami folding (Hypothesis 1). On a more general level, theories linking motor control to learning [cf. 12] can be taken to suggest that load effects on change in performance should align with load effects on performance. Accordingly, we hypothesized that effects of load on learning (change in performance) should also influence performance of Origami folding (Hypothesis 2).

### Materials and methods

In the experimental setting, speed and accuracy of responses were recorded to examine the behavioral changes of the Origami folding task (Task 1) and the secondary tasks (Task 2). Program, data and folding instructions are available online [65, https://osf.io/p3tyf/].

## Participant

Sixty-one participants were recruited for the experiment. Of these, data from eight participants were excluded because they did not complete the experiment. Data from 53 participants are reported ($M_{age}$ = 34.5 years, *SD* = 8.9, *range* = 18 to 58; 29 females, 21 males and 3 others). The mean age of participants was higher than in many laboratory studies in cognitive psychology as students at University of Hagen (state-run distance teaching university in Germany) are older and more heterogeneous in age than students at other universities. All participants had normal or corrected-to-normal vision acuity and hearing ability. Ethics approval was obtained from the Ethics Committee of the German Psychological Association (April 19, 2018). Participants gave their written informed consent before their participation. The experiment was part of the four theses for a Bachelor of Science program. Participants took part in the two-hour experiment voluntarily for no extra reward. Each participant folded 20 Origami figures (five figures four times each). Overall participants folded 1,060 Origami figures.

## Experimental material

**Origami folding task (Task 1).**   Each trial consisted of an Origami folding task (Task 1) and a secondary task (or single-tasking; see Table 1). For the Origami folding task, we selected randomly five relatively easy-to-fold figures: chair, box, penguin, butterfly and frog [see 65]. The Origami chair contained nine steps and all the other Origami figures contained 10 steps. We designed the folding materials in black and white (800 × 800 pixel resolution) by using the software Excel and PowerPoint from Microsoft Office. The order of the Origami folding task was identical for each participant: chair, box, penguin, butterfly, and frog.

**Secondary task variants (Task 2).**   Each participant performed four folding runs with each Origami figure with the same variant of dual-task demand. Each participant was to fold all five Origami figures and received all five variants of dual-task demands (pairing counterbalanced across participants with a Latin square).

*Cognitive verbal secondary task*. The first type was an Origami folding task combined with a cognitive verbal secondary task [6, 52, 66]. On each trial, the participant received a list with three letters (e.g., R M D) and should keep them in mind (see Condition 1 in Fig 1). Letter lists were generated randomly from a letter pool with 16 uppercase consonants: B, C, D, F, G, H, J, K, L, M, N, P, R, S, T, and W. No letter was repeated within a letter list. No vowels were used to minimize the chances of creating a pronounceable string of letters. The letters were 100 pixel high. The letter string was visible for 3 seconds. Then an instruction of an Origami folding step appeared. After the participant folded the paper, she or he pressed the *Enter* key. For

**Table 1. The sequence of the Origami folding task (Task 1) and the secondary tasks.**

|   | Chair | Box | Penguin | Butterfly | Frog |
|---|-------|-----|---------|-----------|------|
| 1 | CV | CS | MI | MN | None |
| 2 | CS | MI | MN | None | CV |
| 3 | MI | MN | None | CV | CS |
| 4 | MN | None | CV | CS | MI |
| 5 | None | CV | CS | MI | MN |

The sequence of Origami folding was the same for each participant (i.e., chair—box—penguin—butterfly—frog). Sequences of the secondary task variants (CV: Cognitive Verbal task; CS: Cognitive Visuospatial task; MI: Motoric Isochronous Tapping; MN: Motoric Non-isochronous Tapping) to the Origami figures were counterbalanced according to a Latin square, shown in the chart. Each row represents the combinations of the two tasks for one participant. The condition within each cell was presented four times per participant (four trials of practice with the same Origami and the same secondary task demand in direct succession).

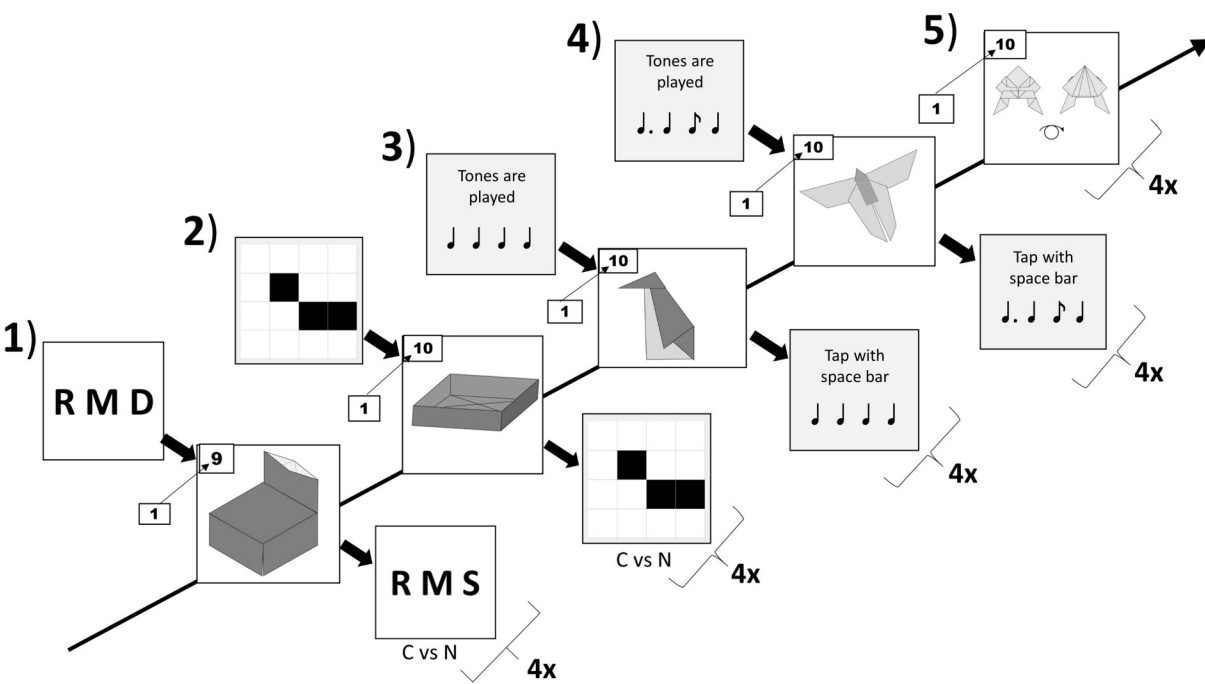

**Fig 1. Schematic depiction of the Origami figures and the secondary tasks.** Each participant folded five Origami figures successively: Chair, box, penguin, butterfly and frog for four times. The secondary tasks were counterbalanced across particicpants. This participant folds the chair figure from folding Step 1 to folding Step 9 in four runs. Each time, the participant receives a list of three letters for three seconds and then an instruction of a folding step. After completing the folding step, the participant verifies whether the second letter list is identical (press *C* key) or different (press *N* key) to the initial list. In the consecutive blocks, the participant folds the box figure (with the cognitive visuospatial memory load), the penguine figure (with motoric isochronous tapping memory load), the butterfly figure (with motoric non-isochronous tapping memory load), and the frog figure (with no memory load).

verification, the participant was shown a second list with three letters. For half of the trials, the letter list was identical to the initial list (true trial). For the other half of the trials, one randomly chosen letter was switched with another letter in the letter pool (false trial). For instance, when the letter list was "R M D" and verification task was "R M D", the answer for the verbal secondary task was "true". If the letter list was "R M D" and the verification task was "R M S", the trial was false. The participant decided whether the second letter list was different (*N* key) or identical to the initial letter list (*C* key).

*Cognitive visuospatial secondary task*. The second type of secondary task variant was to fold the Origami with a visuospatial interference task [6, 67]. The participant was first shown a 4 × 4 grid containing three black squares for 3 seconds (see Condition 2 in Fig 1). Each square was 100 pixel and the grid was 400 pixel in width and in height. The task was to memorize the pattern of squares for later verification. The positions of the squares were generated randomly. The only constraint was that the three squares could not fall in a straight line. In-between new memory items, the participant received an instruction to fold an Origami step. After the folding step was completed, she or he pressed *Enter*. For verification, participants were presented a second grid containing three squares. Trials were constructed so that half of them were true (first matrix identical to second matrix) and half were false. For the false trials, one randomly chosen square was moved by one space in the grid. The direction of movement of this square was generated randomly and was constrained in a way that a straight line was not formed. As shown in Fig 1, if the grid with three black squares was identical before and after the folding

task, participants were to respond "true". The participant pressed the *C* key for true trials and the *N* key for false trials.

*Motoric isochronous tapping secondary task*. The third type was to complete an Origami folding task with a motoric isochronous tapping secondary task. That is, participants were asked to tap repeatedly IOIs (inter-stimulus onset intervals) of equal duration. The participant first listened to an isochronous pacing beat with four tones (each in 440 Hz with a pitch of A4 with a duration of 100ms). The IOI was generated randomly one out of three intervals of one, duple and triple multiples of the basic interval onset 300ms (i.e., 300ms, 600ms, 900ms), similar to metrically structured patterns in other studies [57, 68]. Unlike the original patterns in the study of Povel and Essens [57], we used only the sub-seconds range, as Holm et al. [5] found the interference of load on tapping in the sub-seconds range rather than in the supra-seconds range. One example of the isochronous beat was 300ms, 300ms, 300ms. The beat was played by the laptop's built-in speaker. After the folding task, the participant tapped the rhythm by pressing the space bar. They could tap with fingers of either hand or both hands. Only if they tapped the space bar four times, the next trial appeared. The program recorded the respective inter-response intervals (IRIs) of the taps. We considered the trial correct, when the variability of IRIs was less than 20% of the IOIs regarding absolute timing or relative timing [cf. 69].

*Motoric non-isochronous tapping secondary task*. The fourth type was to pair the Origami folding task with a motoric non-isochronous tapping secondary task. That is, participants tapped the IOIs of arbitrary durations. The participant first listened to a pacing beat with four tones (440 Hz with a duration of 100ms) in unequal intervals. The IOIs were generated randomly from three time intervals (300ms, 600ms, and 900ms). One constraint was that one of the three time intervals could not be chosen for three times to avoid producing the isochronous beat. One possible non-isochronous rhythm was 600ms, 300ms, 300ms. After the folding task, the participant tapped the beats by pressing the space bar. Only if they tapped the space bar for four times, the next trial appeared. The program recorded the IRIs of the taps. The trial was classified as correct, when the variability of IRIs was less than 20% of the IOIs regarding absolute timing or relative timing.

*None*. The last variant was to fold each origami figures for four times without the secondary task.

## Apparatus

The experiment was implemented in Lazarus [70]. The experiment was displayed on a Lenovo Thinkpad T530 laptop with a 12.5-inch display. The keys for entering secondary task responses (*C*, or *N*, or space bar) and folding step ends (*Enter*) were highlighted with colored self-adhesive dots. Additionally, 24 sheets of $210 \times 210$ mm white papers were available for each participant for the Origami folding task.

## Procedure

The experiment was conducted individually in a quiet environment. At the beginning, participants were informed about the procedure of the study and that they could quit the experiment at any time. They signed the form of consent. Data were acquired anonymously, the program registered the demographic information such as age and gender and the folding times and responses to the secondary tasks. The experimenters recorded the accuracy of the folding task. Before folding, the folding instruction handouts were passed to the participant. Participants were informed about the terminologies and meanings of symbols before folding. For instance, they saw the *folding states* in Fig 2, which are snapshots of *folding motions* (indicated by arrows). There are many *creases*, which are line segments on a piece of paper

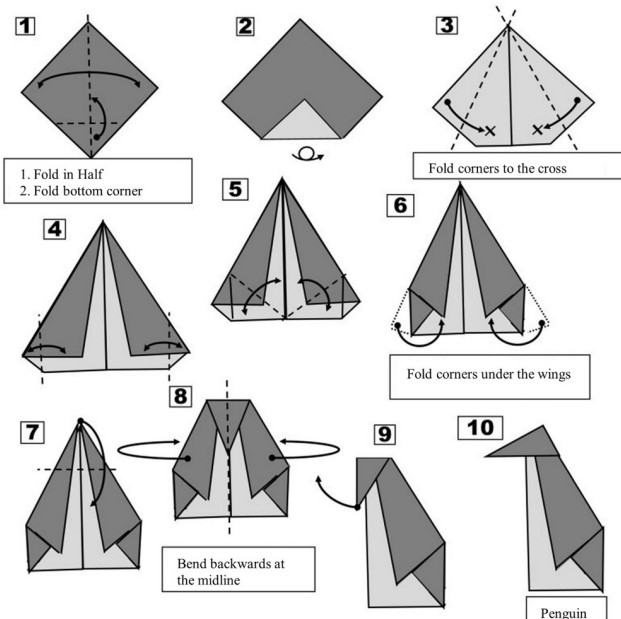

**Fig 2. The folding instruction handout for Origami figure penguin.** Only the critical steps were described (originally in German). In the experiment, participants received each time only one folding step.

[cf. 15]. They can be folded in two ways. A *mountain fold* displayed in solid lines forms a protruding ridge. A *valley fold* shown by dash lines forms an indented trough. The light grey side represents the front side of the paper and the dark grey side represents the backside of the paper. The five-page instructions including all folding steps described the particularly difficult folding steps with textual explanations, which were supposed to help participants to better understand the instructions. Participants did not fold at this stage and only read the folding instructions.

After these general instructions, participants received the papers to fold and the experiment started. To enable time measurement for the folding steps, the participants were only allowed to perform folding steps if indicated by the program. The participants were informed to perform the tasks as accurately and as quickly as possible. During the experiment, Origami folding was only interrupted for introducing the secondary tasks. Before each folding step, the secondary task was presented. The cognitive verbal and cognitive visuospatial secondary tasks were displayed for three seconds. The motoric isochronous tapping and non-isochronous tapping secondary tasks were played with three IOIs (300ms, 600ms, 900ms). The display changed automatically to the folding step instructions after presentation of the secondary task. Once the participants completed the folding step, they pressed the Enter key to proceed. This was also the case if the participants decided that they were unable to complete the step successfully. The experimenter registered the success of each folding step by observation. After the folding step, the secondary task response was registered. No feedback was provided about speed and accuracy.

If an Origami folding run contained wrongly executed steps or was entirely unsuccessful, a hint was given by the experimenter on how to avoid the errors in the following trials before it started. These specific instructions that went beyond a pure pictorial and textual representation enabled the participants to master the steps at which they failed before [see 71]. The support after errors was necessary as we compared the change in performance through practice in

Origami folding. Without the support, many participants would not have completed folding the figures for four times. The entire experiment contained five blocks (each block contained an Origami figure) of four folding trials each. Trial 1 referred to the first run of folding a figure with all folding steps. It is important to note that the secondary task and the Origami figures were unchanged within a block. After self-paced folding of each Origami figure, participant could choose to have a break. Each participant was to fold the five different Origami figures four times each, totalling 20 figures. The overall experiment took around 2 hours per participant.

### Measurements

We recorded the RTs (reaction times) and error rate per folding step for different secondary tasks for each participant. RTs per folding step were determined based on the interval between the onset of the displayed folding step and pressing the *Enter* key to indicate completion. Error rates of each folding step were noted by the experimenters. RTs of the cognitive secondary tasks were based on the interval between the start of the verification task and the registration of the response key (either C or N). Responses of the cognitive secondary tasks and IRIs of the taps were registered by the program as well. Mean RTs and error rates were calculated by using a statistical software R.

### Results

We excluded the last step in all the Origami figures due to a programming error. In addition, we excluded trials with RTs of Origami folding task longer than 100 sec (2.1% of all trials). There was no speed-accuracy trade-off. Rather, we obtained a small and positive ($r = .26$) correlation between error rate and completion time. In detail, we computed a correlation for the RT and the completion time of the five Origami figures individually for each participant and then aggregated across participants.

Given that folding an Origami consisted of different steps, we also explored the variability of time demands in each step of different Origami figures. The average folding time (averaging across all trials) was 19.70 sec ($SD = 7.63$ sec). The average error rate was 10.9% ($SD = 31.1%$). The chair took 19.50 sec ($SD = 15.80$ sec) to be folded and it had an average error rate of 8.7% ($SD_{error} = 28.2%$); box 19.46 sec ($SD = 17.10$ sec; $M_{error} = 11.5%$, $SD_{error} = 31.9%$), penguin 21.24 sec ($SD = 19.17$ sec; $M_{error} = 12.9%$, $SD_{error} = 33.5%$), butterfly 19.19 sec ($SD = 16.87$ sec; $M_{error} = 11.1%$, $SD_{error} = 31.4%$) and frog 18.23 sec ($SD = 15.03$ sec; $M_{error} = 9.9%$, $SD_{error} = 30.0%$). The fluctuation in Fig 3 (error rates see S1 Table in S1 Appendix) likely reflects the differences in complexity among individual folding steps. Taking the most difficult Penguin figure as an example (see Fig 2), the simpler folding Step 2 ("Turn the paper over and fold the lower corner to the middle") required only 4.49 seconds ($SD = 3.33$ sec) on average with an average error rate of 5.66% ($SD_{error} = 23.3%$) in Trial 1. In contrast, Step 6 ("Fold the corners under the inside part of the wings") took 42.31 seconds ($SD = 28.03$ sec). It had an average error rate of 73.58% ($SD_{error} = 44.5%$) in Trial 1. This high variability suggests to proceed by analyzing the effect of dual-task variant on performance and change in performance on the level of entire Origami figures rather than at the level of steps. To avoid the influence of various levels of difficulty caused by different shapes, we used the design of a Latin-square table (see Table 1), so that the different shapes were combined with all kinds of memory load.

In the main analysis, we examined the performance in the Origami folding task by pairing the baseline (no dual-task load) with each dual-task condition (see Table 2). Before coming to the main analysis, we checked whether there were differences among the dual-task conditions and whether there was an effect of practice. A repeated-measures analysis of variance

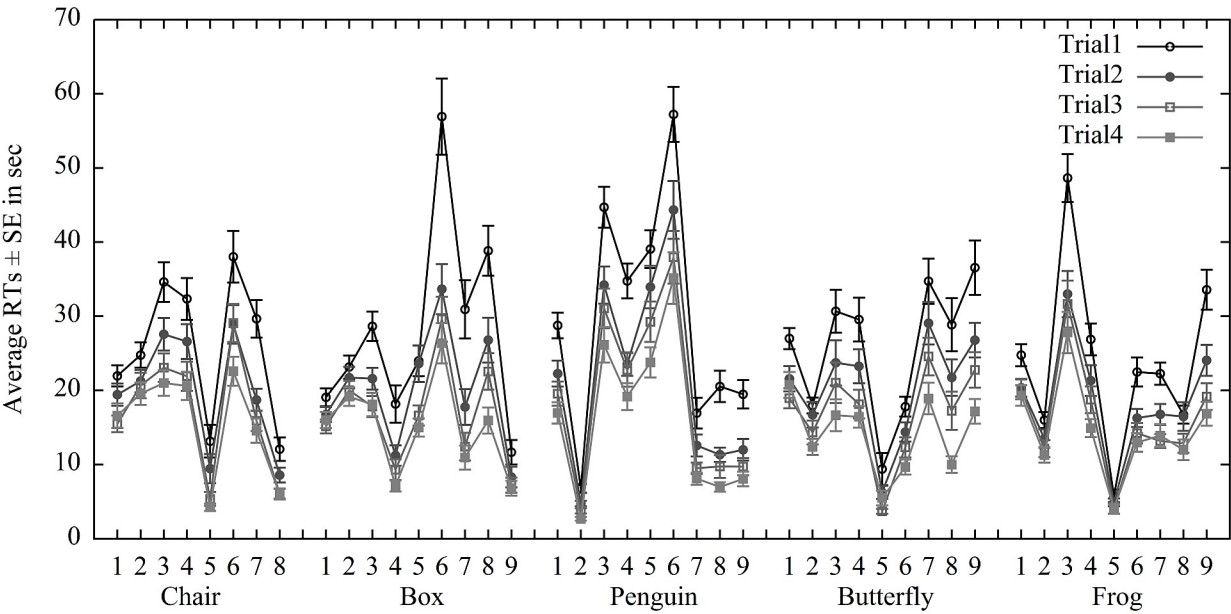

**Fig 3. RTs in four practice trials for individual Origami figures.** The numbers represent the folding steps. Due to a programming error, the last step in all Origami figures was excluded. The error bars reflect the standard errors of the mean.

(ANOVA) was conducted with four variants of memory load (cognitive verbal, cognitive visuospatial, motoric isochronous tapping, motoric non-isochronous tapping) × 4 levels of practice (Trial 1 to Trial 4). The results showed a main effect of memory load, $F(3, 156) = 4.07$, $p = .008$, $\eta_p^2 = .07$, indicating differences among the memory load variants. The main effect of practice, $F(1.84, 95.87) = 41.57$, $p < .001$, $\eta_p^2 = .77$, reflected strong performance gains (here and elsewhere we applied Greenhouse Geisser-correction when appropriate). However, no interaction of memory load and practice was present, $F(5.56, 289.07) = 1.16$, $p = .33$, $\eta_p^2 = .02$.

We aimed at comparing how different variants of memory load affect performance in Origami folding and acquisition of Origami folding (change in performance across practice trials on a specific figure). As learning can be estimated by improvement of performance through

**Table 2. The average RTs and error rates of Origami folding task (T1) with all variants of memory load.**

|  | Trial 1 | Trial 2 | Trial 3 | Trial 4 | Average | Diff_Trial1-4 |
|---|---|---|---|---|---|---|
| | RTs in sec Mean (SD) | | | | | |
| Cog. Verbal | 26.24 (10.36) | 19.08 (7.47) | 15.96 (6.57) | 14.16 (5.79) | 18.86 (8.97) | 12.8 (8.58) |
| Cog. Visuospatial | 24.89 (10.39) | 20.82 (9.74) | 17.66 (8.60) | 15.85 (7.41) | 19.81 (9.67) | 9.04 (8.21) |
| Mot. Iso. Tapping | 27.79 (8.99) | 22.58 (7.85) | 20.04 (9.07) | 17.75 (7.88) | 22.04 (9.20) | 10.40 (6.58) |
| Mot. Noniso. Tapping | 25.63 (10.28) | 19.25 (9.33) | 17.22 (10.06) | 15.25 (8.49) | 19.34 (10.27) | 10.37 (9.22) |
| None | 26.37 (10.81) | 19.09 (8.06) | 15.56 (7.67) | 12.95 (6.18) | 18.49 (9.71) | 13.42 (8.68) |
| | Error rates in % Mean (SD) | | | | | |
| Cog. Verbal | 14.2% (19.7%) | 3.9% (8.3%) | 1.7% (6.7%) | 1.0 (3.9%) | 5.2% (12.5%) | 13.2% (20.0%) |
| Cog. Visuospatial | 21.0% (24.2%) | 13.8% (20.5%) | 6.9% (13.2%) | 7.2% (15.0%) | 12.2% (19.5%) | 13.9% (19.7%) |
| Mot. Iso. Tapping | 17.2% (23.5%) | 11.3% (17.4%) | 8.4% (13.5%) | 6.1% (11.3%) | 10.8% (17.4%) | 11.1% (17.9%) |
| Mot. Noniso. Tapping | 16.5% (20.6%) | 13.1% (17.5%) | 10.7% (19.6%) | 5.9% (11.6%) | 11.5% (18.0%) | 10.5% (16.9%) |
| None | 17.8% (24.3%) | 5.6% (12.9%) | 2.1% (6.9%) | 1.5% (9.3%) | 6.7% (16.2%) | 16.3% (26.6%) |

practice, the interaction of load type and practice is in the focus of the report. The performance of the secondary tasks is reported in the S2 Table in S1 Appendix.

## Origami folding with cognitive visuospatial memory load vs. no load

We used a 2 load types (cognitive visuospatial memory load vs. no load) × 4 levels of practice (Trial 1 to Trial 4) repeated-measures ANOVA to analyze the *RTs*. The results revealed an interaction of load type and practice, $F(1.88, 97.86) = 3.99$, $p = .02$, $\eta_p^2 = .07$. It indicated that the gain in performance across practice was larger for single-tasking (RTs: $\Delta M = 13.42$ sec; $\Delta SD = 8.68$ sec) than for visuospatial dual-tasking ($\Delta M = 9.04$ sec; $\Delta SD = 8.21$ sec) as shown in Fig 4A. Furthermore, there was a main effect of practice, $F(1.93, 100.34) = 104.71$, $p < .001$, $\eta_p^2 = .67$, but no significant effect of load type, $F(1, 52) = 1.33$, $p = .25$, $\eta_p^2 = .03$.

The separate ANOVA on *error rate* showed no interaction of load type × practice, $F(2.42, 125.89) = 0.70$, $p = .53$, $\eta_p^2 = .01$, but a main effect of load type, $F(1, 52) = 8.14$, $p = .006$, $\eta_p^2 = .14$, suggesting participants made more errors when folding Origami with the cognitive visuospatial secondary task than when folding Origami alone (see Fig 4B). The main effect of practice again showed the strong training effect, $F(1.90, 98.99) = 28.39$, $p < .001$, $\eta_p^2 = .35$.

## Origami folding with cognitive verbal memory load vs. no load

We analyzed *RTs* in the Origami folding task with a 2 load types (cognitive verbal memory load vs. no load) × 4 levels of practice repeated-measures ANOVA. We found neither a main effect of load type nor an interaction effect of load type × practice, $Fs < 1$. Only a main effect of practice was revealed, $F(1.61, 83.70) = 118.33$, $p < .001$, $\eta_p^2 = .70$. Similarly, the ANOVA on *error rate* did not show any effect of load type and any interaction of load type × practice, $Fs < 1.38$, but only a main effect of practice, $F(1.50, 77.83) = 23.44$, $p < .001$, $\eta_p^2 = .31$. The

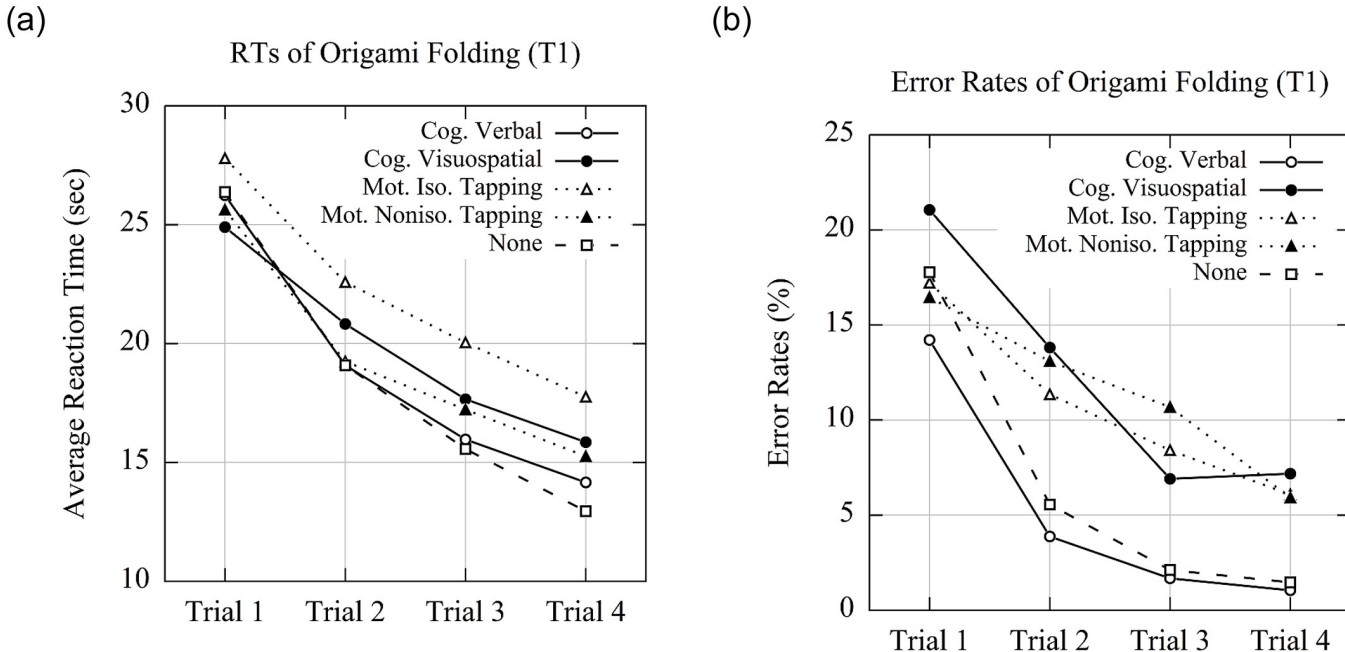

**Fig 4. Performance of Origami folding task.** (A) RTs and (B) error rates of Origami folding task (Task 1) when paired with different variants of the memory load (i.e., cognitive verbal, cognitive visuospatial, motoric isochronous tapping, motoric non-isochronous tapping, and none).

results did not suggest any significant difference in Origami folding with a concurrent verbal memory load compared to without memory load.

## Origami folding with motoric isochronous tapping memory load vs. no load

We analyzed *RTs* in the Origami folding task in a 2 load types (motoric isochronous tapping memory load vs. no load) × 4 levels of practice repeated-measures ANOVA. We found an interaction effect of load type × practice, $F(1.90, 98.97) = 67.23$, $p < .001$, $\eta_p^2 = .56$, suggesting RTs were reduced more in the single-task condition across the four practice trials than in the motoric isochronous tapping dual-tasking condition (RTs: $\Delta M = 10.37$ sec; $\Delta SD = 9.26$ sec). Furthermore, there were main effects of load type, $F(1, 52) = 328.76$, $p < .001$, $\eta_p^2 = .86$, and of practice, $F(1.91, 99.33) = 65.37$, $p < .001$, $\eta_p^2 = .56$.

The ANOVA on *error rates* for the Origami folding task revealed no interaction of load type × practice, $F(2.36, 122.94) = 2.69$, $p = .06$, $\eta_p^2 = .05$, but main effects of load type, $F(1, 52) = 4.60$, $p = .04$, $\eta_p^2 = .08$, and practice, $F(1.71, 89.10) = 20.76$, $p < .001$, $\eta_p^2 = .29$.

## Origami folding with motoric non-isochronous tapping memory load vs. no load

The ANOVA on *RTs* with 2 load types (motoric non-isochronous tapping memory load vs. no load) × 4 levels of practice repeated-measures ANOVA revealed only a main effect of practice, $F(2.06, 107.19) = 103.69$, $p < .001$, $\eta_p^2 = .67$. No other effect was found, load type and load type × practice, $Fs < 1.76$. The estimated Bayes factor (alternative/null) for the interaction suggested that the data were .056:1 in favor of the null hypothesis, or rather, 17.84 times more likely to occur under the model excluding an effect for load type and practice, rather than the model with it. The ANOVA on *error rate* showed the main effects of load type, $F(1, 52) = 6.50$, $p = .01$, $\eta_p^2 = .11$, and practice, $F(2.22, 115.24) = 21.53$, $p < .001$, $\eta_p^2 = .29$, as well as the interaction effect, $F(1.98, 103.03) = 3.21$, $p = .05$, $\eta_p^2 = .06$. It indicated that participants made more errors when performing and learning Origami folding with concurrent isochronous tapping secondary task.

## Discussion

The main purpose of this study was to compare how different variants of memory load affect performance in Origami folding and acquisition of Origami folding (change in performance through practice on a specific figure). The experiment reported here therefore concerned the interaction of load type and practice. Such an interaction was obtained when comparing Origami folding with (1) the cognitive visuospatial memory load vs. no memory load as well as when comparing (2) motoric isochronous tapping memory load with the no memory load baseline, which confirmed Hypothesis 1. In line with theories suggesting that learning operates on the level that (at a given point in practice) demands the most control [cf. 12], we did not observe cases where a dual-task variant influenced performance while it did not affect learning (i.e. change in performance), and thus confirmed Hypothesis 2.

### Visuospatial resource

Based on Baddeley [28], a dual-task paradigm with a memory load task was employed to test whether the execution vs. acquisition of Origami folding shared resources in the phonological loop or in visuospatial sketchpad. Indeed, an interaction of load type and practice (RTs) was revealed when participants performed Origami folding with cognitive *visuospatial* secondary

task. It suggests a decremental effect in visuospatial sketchpad on acqusition rather than on execution of Origami folding when paired with visuospatial secondary task. One potential explanation is that the cognitive process of Origami folding involves the visuospatial sketchpad and taxing this representation via a secondary task compromises control-based learning [cf. 12]. During the process of converting the two-dimensional instructions to a three-dimensional object, many spatial concepts are involved, such as in mountain folding, front side of the paper, creases, and motion arrows [15]. Moreover, evidence has shown that sequential skill learning relies essentially on error detection and error correction [72]. While spatial working memory is demanded for processing motor errors and updating motor control [73], the visuospatial resources in the visuospatial sketchpad are needed for sequential skill learning.

In contrast, the cognitive verbal secondary task showed neither a main effect of load type nor an interaction of load type and practice. Previous studies on implicit sequence learning have shown that an auditory (auditory-verbal) task can significantly interfere with the acquisition of a sequential skill in a spatial (visual-manual—screen locations to key locations) task [29, 30]. Moreover, Tenbrink and Taylor [13] stated that while folding participants reformulate and reconceptualize the instruction in their own words or thoughts to conduct the movements. The pictorial information can be transformed into code held in the phonological loop to ease the load in the pictorial channel by inference-making [cf. 74, 75]. Presumably, after seeing the picture of one folding step, some participants used their own words explaining which action they should perform. However, in the current study, we did not observe detrimental effects of verbal memory load on sequence learning task. Potentially, our taxing of memory load in the verbal task was only mild. We required participants to recognize the change of one letter in a list of three letters. Further studies should be conducted with more challenging verbal secondary tasks.

## Timing resource

Based on the Multiple Resource Theory [48], our dual-task paradigm examined potential resource conflicts of cognitive vs. motor control in Origami folding. Tapping should tax motor control. Indeed, the analyses of Origami completion times showed a main effect of load type and an interaction effect of load type and practice in Origami folding with isochronous tapping secondary task. In contrast, neither a main effect nor an interaction was revealed when Origami folding was paired with non-isochronous tapping secondary tasks. It suggested only the impairment of executing as well as learning Origami folding, while holding the isochronous rhythmic patterns in mind. According to Povel [55], participants attempt to perceive and estimate the rhythmic tempi in internal clock with isochronous temporal patterns. Yet when reproducing non-isochronous rhythmic patterns, the internal clock cannot be generated and people tend to resort to the organizing principles of assimilation and distinction [57, 61, 76]. They assimilate similar temporal patterns toward 1:1 ratio. When the temporal patterns distinguish like 2:3, they tend to categorize the patterns into long/short ratios. For instance, rhythmic patterns with 3:2:2 ratio (e.g., 900ms, 600ms, and 600ms) tend to be reproduced with a long/short/short ratio of 2:1:1. In short, it is conceivable that absolute timing mechanism were activated when participants tapped isochronous patterns. The relative timing mechanism were activated when participants tapped non-isochronous patterns. The analyses on error rates of the tapping secondary task in S2 Table in S1 Appendix supported the argument that participants employed different timing strategies in tapping tasks. When considering absolute timing (self-produced IRIs in variance of 20% of the absolute IOIs) as correct trials, participants made much more errors in the motoric non-isochronous tapping secondary task compared to the motoric isochronous tapping secondary task (90.9% vs. 61.6%). In contrast, when considering

relative timing (self-produced IRIs in variance of 20% of the relative IOI ratios), error rates did not show any difference between the isochronous vs. non-isochronous tapping task (64.3% vs. 54.3%). Isochronous rhythmic patterns can be perceived, memorized and reproduced via absolute timing mechanism, whereas non-isochronous rhythmic patterns via relative timing mechanism. As sequential skills involve accurate timing in movement [12], it might be the case that the absolute timing mechanism in the internal clock is needed for executing and learning of Origami folding. When performing Origami folding with isochronous tapping, the absolute timing mechanism is overload, leading to impairments in execution and acquisition of Origami folding.

## Acquisition vs. performance in origami folding

Taken together, this study did not show a general memory load effect shared across all types of dual-tasking, but rather indicated load effects for specific conditions on folding performance and its change with practice. Interestingly, a concurrent visuospatial secondary task had detrimental effects on learning rather than on performing Origami folding. In line with the previous studies [7, 8, 14], selective memory load does not necessarily affect execution vs. acquisition of sequential skills to the same extent. Yet, in line with theories suggesting that learning operates on the level that (at a given point in practice) demands the most control [cf. 12], there was no case of a memory load leading to a main effect of load, but no interaction of load and practice in folding time.

We studied the acquisition of a sequential skill under memory load and observed strong training effects in Origami folding task across practice trials. Accordingly, memory should have been taxed less and less across trials by the Origami folding task. To the extent that participants became more proficient in the primary task, more memory resources should have been left for the secondary task. However, according changes in performance in the secondary task were only observed for the cognitive verbal secondary task, which showed a strong improvement from Trial 1 to Trial 4, although participants received different letter triplets each time (see S2 Table in S1 Appendix). Given that this dual-task variant had no impact on Origami folding, this pattern might reflect quick practice gains in this secondary task rather than that resources freed in the primary task (Origami folding) were used for the verbal secondary task.

Using Origami folding as a task, the study tested the resources that are relevant to learning a motor task through visualisation. Comparing the impact of different dual-task variants on practice gains in folding suggested that resources involved in isochronous timing and in visual imagery were indeed relevant. Given that we studied the early part of skill acquisition (rather than automatization of motor patterns across many sessions of pratice), it seems reasonable that task performance and learning were mainly guided by visual representations. In line with the theory of Willingham on the connection of learning and motor control [12], participants might mainly rely on and learn based on visual representations early in practice, which is within the scope of our experiment. In work on sequence learning [77], it has been shown that participants in the early phase of practice learn the sequence based on the positions of targets and responses on screen and response panel rather than sequence of finger movements. When practice extends across many sessions, finger movements might become more relevant. Accordingly, a visual-spatial secondary task should be potentially relevant to interfere with the encoding of the sequence or retrieving of the encoded sequence information.

As we aimed to study the early parts of acquisition of Origami folding, we tested novices as they (due to the small amount of prior knowledge) should show the largest learning gains and they should be affected the most by taxing the cognitive resources potentially relevant for

acquiring the sequential skill. Using a within-subject design, we avoided that between subjects differences might confound the results. By testing participants with expertise, the potential to find out which resources are relevant for acquiring the skill (by performing different secondary tasks) might have been more limited, as they might rely on many (sub)processes that are automatized and are no longer affected by taxing the resources by a secondary task. In future studies, data from longer practice phases in novices might be complemented with expert performance to pin down the extent to which cognitive resources that are necessary to control the hierarchical skill of Origami folding are subject to practice-related change.

In line with previous studies dealing with acquisition of sequential skill through using Origami [16–18], the current study suggests that Origami folding can be a potential task to measure sequence learning. First, it shares common characters of sequential skills, which cosists of series of actions following certain sequences. Participants receive step-by-step instructions and they should read, reformulate, reconceptualize and evaluate while folding [cf. 13]. This can show in time demands. For instance, the longest folding time was observed for Step 6 of Penguin (see Fig 3). The instruction for this step was to "fold the corners under the inside part of the wings". Participants should fold the corners first inside and then put them under the wings. This involved motor imagery to off-line simulate the action and execute the action. The illustration suggests that Origami folding is thus much more challenging compared to other lab tasks on sequence learning [30, 31, 78, 79]. Second, although many people have basic experience in folding cranes or planes, we still observed substantial improvements of Origami folding between Trial 1 and 4 (see Fig 4). It suggests that Origami folding can be used to measure the acquisition of sequential skills.

## Limitations

There are several limitations of the study. During the experiment, participants who folded the Origami wrongly were given more time and a chance to learn and practice again. One can argue that by this we took a conservative approach and potentially underestimate the learning effects (i.e., difference between Trial 1 and 4), as we helped participants to overcome large difficulties encountered early on. However, the support after errors was necessary to secure that participants would be able to complete four trials per Origami figure (rather than that some participants would give up within the first trial of a particular figure). Providing a chance to correct errors and receive support should ease frustration and motivate participants in the two-hour experiment. Furthermore, the folding difficulty among steps within the Origami figures was of high variability, leading to variance in folding time. Moreover, prior knowledge and spatial ability play important roles in performing and learning Origami folding. In the study, participants were instructed with pictures of folding steps without textual descriptions. We observed that some participants failed to understand the folding steps and behaved aggressively after the experiment. A study on comics reading [80] documented that inexperienced comic readers had difficulty in reading picture-only comics compared to text-picture comics. Further research should be conducted to explore the influence of prior knowledge on Origami folding on comprehending text-picture instructions.

More studies are needed for replication and generalization of the load effects obtained. Further studies should use a complimentary research strategy to dissociate memory load effects on practice vs. on performance by comparing three conditions. (A) a group practices with no memory load and is tested with memory load. (B) a group practices with memory load and is tested with memory load. (C) a group practices and is tested without memory load. The drawback of this strategy would be that participants in the test phase in part need to accustom to the novel situation of memory load being present, which might be an extra factor to consider.

In addition, a study with (much) more than 4 trials per Origami would be useful to estimate asymptotes, as we observed still improved performance in Trial 4 compared to Trial 3 (S3 Table in S1 Appendix). Further studies should also investigate whether showing the participants a video with hand movements while folding instead of static graphs can influence the acquisition of sequential skills under memory load, as animated instructional materials for Origami folding can lead to better learning outcome than static materials [cf. 17]. Perceiving information with human movement can activate the Mirror Neuron System, which seems essential to emulate the sequential skills [20, 21]. Besides, participants may learn the Origami folding task in a more fluent manner with animated videos compared to stage-based static graphics [81].

## Conclusion

The study presented here suggests that the acquisition of sequential skills with practice can be studied by using a real-life activity. Dual-task manipulations suggested that participants do not depend upon verbal resources when folding Origami, but seem to rely on visuospatial codes and timing for controlling performance and for learning.

## Supporting information

**S1 Appendix.**
(DOCX)

## Acknowledgments

We thank Robert Szwarc for the support in programming the software and making the figures.

## Author Contributions

**Conceptualization:** Fang Zhao, Robert Gaschler.

**Data curation:** Fang Zhao.

**Formal analysis:** Fang Zhao.

**Investigation:** Fang Zhao, Anneli Kneschke, Simon Radler, Melanie Gausmann, Christina Duttine.

**Methodology:** Fang Zhao.

**Project administration:** Fang Zhao.

**Resources:** Fang Zhao, Anneli Kneschke, Simon Radler, Melanie Gausmann, Christina Duttine.

**Software:** Fang Zhao.

**Supervision:** Fang Zhao, Robert Gaschler.

**Validation:** Fang Zhao.

**Visualization:** Fang Zhao.

**Writing – original draft:** Fang Zhao.

**Writing – review & editing:** Fang Zhao, Robert Gaschler, Hilde Haider.

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
