## [Decision Letter · Decision Letter 0]

27 May 2020

PONE-D-20-08555

Origami folding: Memory load effects on acquisition of sequential skills

PLOS ONE

Dear Dr. Zhao,

Thank you for submitting your manuscript to PLOS ONE. After careful consideration, we feel that it has merit but does not fully meet PLOS ONE’s publication criteria as it currently stands. Both reviewers agreed that the study is interesting and timely, but that it needs some polishing before being published. Therefore, we invite you to submit a revised version of the manuscript that addresses the points raised during the review process.

We look forward to receiving your revised manuscript.

Kind regards,

Juan Cristobal Castro-Alonso, Ph.D.

Academic Editor

PLOS ONE

Journal Requirements

Reviewers' comments:

Reviewer's Responses to Questions

**Comments to the Author**

1. Is the manuscript technically sound, and do the data support the conclusions?

Reviewer #1: Partly

Reviewer #2: Partly

2. Has the statistical analysis been performed appropriately and rigorously? 

Reviewer #1: No

Reviewer #2: Yes

3. Have the authors made all data underlying the findings in their manuscript fully available?

Reviewer #1: Yes

Reviewer #2: No

4. Is the manuscript presented in an intelligible fashion and written in standard English?

Reviewer #1: Yes

Reviewer #2: Yes

5. Review Comments to the Author

Reviewer #1: Review the manuscript titled as “Origami folding: Memory load effects on acquisition of sequential skills” (PONE-D-20-08555)

I find this study is interesting.

Major issues

1. Using within-subject design, the first trial, participants were novices, but for the 2nd or 3rd and the 4th trials, whether participants have been proficient for the primary task then more memory resources have left for secondary task? Therefore, the secondary task for the 2nd , the 3rd and the 4th is not under the same situation of the 1st trial.

2. Authors may cite some previous paper folding studies dealing with the acquisition of sequential skills, any papers there or this study is the first study?

3. Any results comparing the folding of box, chair, frog etc.?

4. Whether folding the different shapes will impose the same level of difficulty?

Minor issues

1. Memory load rather than memory loads --- line 49

2. One folding step rather than a folding step --- line 109

3. Any hypotheses before conducting the experiment? --- Research Question section

Reviewer #2: Thank you for the submission. The design of the study is interesting, however, I am very confused with the aim of the study. The major reason might be that the research question and conclusion are not coherent with the methodology as well as the analysis.

In the research question section, the aim was made explicitly on “using dual-task paradigm to dissociate the memory load on execution from acquisition”. One would expect to see comparison between execution and acquisition. However, the actual analyses were on interaction effects between secondary task and number of trials. To be exact, the author conducted repeated measure separately on various the secondary tasks comparing the effect between with and without the secondary task on an individual students’ completion time across different tasks (yet what is 4 repetition in here?). Learning time or performance is not reported at all. The author stated “There was no speed-accuracy tradeoff” but it doesn’t mean that the completion time is equivalent to accuracy. Also, as the author described on page 16, RT is related to the task complexity. Of course there is a main task effect in the result. Later in the conclusion, the author stated “main effect of secondary task type (indicating performance)” which added even more confusion… so what are being tested?

In addition to that, Line 72-73 & 77-78, added more confusion in understanding what is being tested in here. In general, how well one acquire a skill is measured by performance. However, the first statement sounds like they are two different things and thus they are being compared in this study. Then in the later statement, the author tried to compare performance with change of performance which is invalid comparison. These two statements are confusing.

Other than the unclear aim, in the procedure, page 15 line 349 – 358, participants who fold the Origami wrongly were given more time and chance to learn and practice again. This is NOT a fair learning condition. Thus I won’t be surprised when the author claimed Line 462-464 “we did not observe cases where a dual task variant influenced performance while it did not affect learning (i.e. change in 464 performance)”.

Also, in the analysis (page 17), when the author wrote “ANOVA …2x4…(results) revealed”, I assumed this was a two-way ANOVA and was about to comment on the abnormal degree of freedom. But soon I saw Greenhouse Geisser-correction was applied, then I realised this is actually a repeated-measures ANOVA, and everything makes a lot more sense. It would be a lot clearer if the author can describe clearly right at the beginning. Furthermore, it is recommended to test the assumption of sphericity before the main analysis.

In the result section, I am not surprised that there is no effect for auditory secondary task. The core of dual-task paradigm is to have a secondary task that take the same working memory resources with the primary task. Learning Origami on the screen is basically a visual task which has nothing to do with auditory loop. Moreover, I do see the value of this study if the aim of the study is to identify which secondary test would potentially affect the learning of origami (which is actually closer to the running head). The author conducted the study with different secondary task, and it’s a shame that the effect of all the secondary task are not compared directly. Yet, in order to achieve this, it would require the author to rewrite (and maybe even re-analysis) the whole article with a clearer (and actually different) aim.

Minor changes:

Line 41 inconsistent referencing style

Line 57, change “showed a increase” to “showed an increase”

Line 57, replace “with” with “when”

Line 68, “vs.”, do you mean “and?”

Line 71, “4 repetitions” or “4 attempts”? The participants folded the shape 4 times, that is 3 repetitions, right? Or the participants could fold the paper while learning but that original attempt didn’t count? Later in table 1 (page 13-14, line 313 ), the participants actually have to fold the same shape 5 times. This is quite confusing here. Please illustrate clearly.

Line 93-109 go into methods

Section “Origami Folding Task” Describe more about the machine of learning origami or sequential motor skills. See reference below.

Line 130, for origami task, it is understandable why visuospatial sketchpad is needed as the person is required to see and read the picture. However it is not unclear why phonological loop is also needed for such a visual task. Please elaborate.

Line 209, 206 “ [54, https://osf.io/p3tyf/]” The web is not working - page not found.

Line 220 “part of 4 Bachelor of Science theses”? part of 4-year Bachelor of Science thesis? part of the 4 theses for a Bachelor of Science programme?

Page 15 line 360: RT ---- PLEASE present both the spelled-out and the short form when the abbreviation first appeared. I keep questioning why the authors keep using completion time as reaction time for the whole time until the discussion which is the end of the manuscript!

Reference:

Castro-Alonso, J. C., Ayres, P., & Paas, F. (2014). Dynamic Visualisations and Motor Skills. In W. Huang (Ed.), Handbook of Human Centric Visualization (pp. 551-580). New York, NY: Springer New York.

Wong, A, Marcus, N, Ayres, P, Smith, L, Cooper, G.A, Paas, G.W.C, & Sweller, J. (2009). Instructional animations can be superior to statics when learning human motor skills. Computers in Human Behavior, 25(2), 339–347. doi:10.1016/j.chb.2008.12.012

Wong, A., Leahy, W., Marcus, N., & Sweller, J. (2012). Cognitive load theory, the transient information effect and e-learning. Learning and Instruction, 22(6), 449-457. doi:10.1016/j.learninstruc.2012.05.004

6. PLOS authors have the option to publish the peer review history of their article (what does this mean?). If published, this will include your full peer review and any attached files.

Reviewer #1: No

Reviewer #2: No

---

## [Author Response · Author response to Decision Letter 0]

7 Jul 2020

Manuscript: PONE-D-20-08555

Title: Origami folding: Memory load effects on acquisition of sequential skills

Journal: PLOS One

Response to Reviewer #1

1. Using within-subject design, the first trial, participants were novices, but for the 2nd or 3rd and the 4th trials, whether participants have been proficient for the primary task then more memory resources have left for secondary task? Therefore, the secondary task for the 2nd, the 3rd and the 4th is not under the same situation of the 1st trial.

Reply: Thank you for pointing this out. We studied acquisition of serial order under memory load and indeed memory load should have less influence the more proficient participants are with a specific Origami figure. We now mention in the Discussion that across trials working memory should be taxed less and less.

2. Authors may cite some previous paper folding studies dealing with the acquisition of sequential skills, any papers there or this study is the first study?

Reply: To our knowledge, our study is the first study using paper folding to examine how sequential skills are acquired through practice. We have mentioned it in the Introduction under the section Origami Folding Task as well as in the Discussion.

3. Any results comparing the folding of box, chair, frog etc.?

Reply: We have added the results (reaction times and error rates) comparing folding each figure in the Result section and Appendix in the manuscript (see Fig 2 and S1 Table in Appendix). 

4. Whether folding the different shapes will impose the same level of difficulty?

Reply: Please see the reply to Comment 3. We have added it to the Results section. Different shapes as well as different steps within a figure can impose different levels of difficulty. The different levels of difficulty might influence the effect of memory load. However, we used the design of Latin-square table, so that the different shapes are combined with all kinds of memory load. 

5. Memory load rather than memory loads --- line 49

Reply: We have corrected it.

6. One folding step rather than a folding step --- line 109

Reply: We have corrected it.

7. Any hypotheses before conducting the experiment? --- Research Question section

Reply: Two hypotheses were added under the Research Question section. 

 

Response to Reviewer #2

1. Thank you for the submission. The design of the study is interesting, however, I am very confused with the aim of the study. The major reason might be that the research question and conclusion are not coherent with the methodology as well as the analysis.

In the research question section, the aim was made explicitly on “using dual-task paradigm to dissociate the memory load on execution from acquisition”. 

Reply: Thank you for pointing this out. We explained the research question now more clearly about and added the hypotheses. We aimed at comparing how different variants of memory load affect performance in Origami folding and change in performance across trials with practice on a specific figure. 

2. One would expect to see comparison between execution and acquisition. 

Reply: We now acknowledge in the Discussion that further studies would use another research strategy to dissociate memory load effects on practice vs. on performance. (A) a group practices with no memory load and is tested with memory load. (B) a group practices with memory load being tested with memory load and. (C) a group practices and is tested without memory load. The drawback of this strategy would be that participants at test phase in part need to accustom to the novel situation of memory load being present, which might be an extra factor to consider.

3. However, the actual analyses were on interaction effects between secondary task and number of trials. To be exact, the author conducted repeated measure separately on various the secondary tasks comparing the effect between with and without the secondary task on an individual students’ completion time across different tasks (yet what is 4 repetition in here?).

Reply: Thank you for pointing out the confusing term. Before the experiment, participants received the folding instruction and did not fold. During the experiment, they folded 4 times. We thus corrected the term “repetition” into “trial”. One trial means one run of folding an Origami figure.

 4. Learning time or performance is not reported at all. 

Reply: The learning time and error rates of Origami-folding task (T1) were reported in Table 2. The learning time or performance of each figure are added to S1 Table in Appendix. 

5. The author stated “There was no speed-accuracy tradeoff” but it doesn’t mean that the completion time is equivalent to accuracy. Also, as the author described on page 16, RT is related to the task complexity. Of course there is a main task effect in the result. Later in the conclusion, the author stated “main effect of secondary task type (indicating performance)” which added even more confusion… so what are being tested?

Reply: Thank you for pointing this out. We tried to further clarify this. The main effect of task is not about the specific Origami figure it is about a kind of memory load being present vs. absent. To avoid the confusion, we have changed “task type” into “load type”.

6. In addition to that, Line 72-73 & 77-78, added more confusion in understanding what is being tested in here. In general, how well one acquire a skill is measured by performance. However, the first statement sounds like they are two different things and thus they are being compared in this study. Then in the later statement, the author tried to compare performance with change of performance which is invalid comparison. These two statements are confusing.

Reply: Thank you for this comment. We tried to further clarify how we can assess skill acquisition. How well one acquires a skill can be estimated by change in performance from Trial 1 to Trial 4. It would be interesting to know how close to the asymptote participants were brought by the 4 numbers of trials that we granted the participants. We acknowledge in the Discussion that a study with (much) more than 4 trials per Origami would be useful to estimate asymptotes. In S3 Table in Appendix, there is a comparison between Trial 1 to Trial 4. As there was still an improvement from Trial 3 to Trial 4, three trials are not sufficient to reach the asymptote.

7. Other than the unclear aim, in the procedure, page 15 line 349 – 358, participants who fold the Origami wrongly were given more time and chance to learn and practice again. This is NOT a fair learning condition. Thus I won’t be surprised when the author claimed Line 462-464 “we did not observe cases where a dual task variant influenced performance while it did not affect learning (i.e. change in 464 performance)”.

Reply: Indeed, one could argue that we take a conservative approach and potentially underestimate the learning effects (i.e., difference between Trial 1 and 4) as we helped participants to overcome large difficulties encountered early on. We now briefly mention in the Discussion that support after errors was necessary to secure that participants would be able to complete four trials per Origami figure (rather than that some participants would give up within the first trial of a particular figure). 

8. Also, in the analysis (page 17), when the author wrote “ANOVA …2x4…(results) revealed”, I assumed this was a two-way ANOVA and was about to comment on the abnormal degree of freedom. But soon I saw Greenhouse Geisser-correction was applied, then I realised this is actually a repeated-measures ANOVA, and everything makes a lot more sense. It would be a lot clearer if the author can describe clearly right at the beginning. Furthermore, it is recommended to test the assumption of sphericity before the main analysis.

Reply: Thank you for pointing this out. We now make our strategy of analysis more explicit upfront.

9. In the result section, I am not surprised that there is no effect for auditory secondary task. The core of dual-task paradigm is to have a secondary task that take the same working memory resources with the primary task. Learning Origami on the screen is basically a visual task which has nothing to do with auditory loop. Moreover, I do see the value of this study if the aim of the study is to identify which secondary test would potentially affect the learning of origami (which is actually closer to the running head). The author conducted the study with different secondary task, and it’s a shame that the effect of all the secondary task are not compared directly. Yet, in order to achieve this, it would require the author to rewrite (and maybe even re-analysis) the whole article with a clearer (and actually different) aim.

Reply: Thank you for the comment. Our aim is to test the effect of memory load on performance and on learning (improvement of performance through practice) of Origami folding. We therefore compared the performance of Origami folding (with memory load vs. without memory load) in 4 trials (Trial 1 to Trial 4 through practice). In order to make the aim clearer, we explained the research aim more explicitly in the manuscript. We added an overall repeated-measures ANOVA with 4 kinds of memory load (without the condition with no memory load) and 4 trials (time course of practice from Trial 1 to 4) at the beginning of the analyses. The significant main effect suggested that there were strong differences among the memory load variants. Given that we use a collection of different WM tasks, we deem an approach combining the paired analysis (no load vs. one specific load) as most appropriate. 

Although learning Origami folding is a visual task, it can still interfere with auditory loop. Previous studies on implicit sequence learning (Roettger et al., 2019; Schumacher & Schwarb, 2009) have shown that an auditory (auditory-verbal) task can significantly interfere with the acquisition of a sequential skill in a spatial (visual manual – screen locations to key locations) task. In addition, Tenbrink and Taylor (2015) mentioned that while folding participants reformulate the instruction in their own words or thoughts and conduct the movements. The integrative model of text-picture integration (Schnotz, 2014) suggests pictorial information can transform into phonological loops to ease the load in the pictorial channel. However, in the current study, we did not observe such detrimental effects (of verbal memory load) on sequence learning task. It can be due to the low taxing of memory load in the verbal task. We asked participants to recognize the change of one letter in a list of 3 letters. Further studies should be conducted with more challenging verbal secondary tasks.

10. Line 41 inconsistent referencing style

Reply: We have corrected it.

11. Line 57, change “showed a increase” to “showed an increase” 

Reply: We have corrected it.

12. Line 57, replace “with” with “when” 

Reply: We have corrected it.

13. Line 68, “vs.”, do you mean “and?”

Reply: We have corrected “vs.” into “and”.

14. Line 71, “4 repetitions” or “4 attempts”? The participants folded the shape 4 times, that is 3 repetitions, right? Or the participants could fold the paper while learning but that original attempt didn’t count? Later in table 1 (page 13-14, line 313), the participants actually have to fold the same shape 5 times. This is quite confusing here. Please illustrate clearly.

Reply: Thank you for pointing the confusion out. We agree that “repetition” is not the best term, as the participants did not fold while learning. They only read the handouts of the folding instructions with all the folding steps. Then they folded 4 times in the experiment. We changed “repetitions” into “trials” and explicitly explained it in the abstract and the manuscript. Trial 1 refers the first run of folding one figure. Trial 2 refers to the second run of folding the same figure.

15. Line 93-109 go into methods

Reply: We have integrated this paragraph into methods.

16. Section “Origami Folding Task” Describe more about the machine of learning origami or sequential motor skills. See reference below.

Reply: We have added the mechanism of learning origami or sequential motor skills to the manuscript. We have mentioned the importance of Mirror Neuron System. Thank you for the papers that were recommended.

17. Line 130, for origami task, it is understandable why visuospatial sketchpad is needed as the person is required to see and read the picture. However it is not unclear why phonological loop is also needed for such a visual task. Please elaborate.

Reply: Thank you for pointing it out. We have explained now more explicitly, why phonological loop can be involved. According to procedure of Origami folding (Tenbrink & Taylor, 2015), people read and reformulate the folding instructions in their own words at first. In accordance with the Integrative Model of Text-Picture Integration (Schnotz, 2014), pictorial information can be transformed from images in pictorial channel into sounds in verbal channel. When the mental representation of the instruction is constructed, the information can be traced back to verbal and pictorial information. One example would be after seeing the picture of one folding step, some participants used their own words explaining which action they should perform.

18. Line 209, 206 “ [54, https://osf.io/p3tyf/]” The web is not working - page not found.

Reply: The website can be opened by the authors and their friends as well as the first reviewer. We have positive test results from Germany, UK and China. Besides, we have used the Open Science Framework repository for several other papers and other reviewers could open the link in this repository. We are thus not sure why the web link could not be opened by Reviewer 2. When necessary, we can send the program, the data and the folding steps (the documents in the repository) to Reviewer 2 by email. 

19. Line 220 “part of 4 Bachelor of Science theses”? part of 4-year Bachelor of Science thesis? part of the 4 theses for a Bachelor of Science programme?

Reply: Thank you for pointing out the confusion. They were part of the 4 theses for a Bachelor of Science program. 

20. Page 15 line 360: RT ---- PLEASE present both the spelled-out and the short form when the abbreviation first appeared. I keep questioning why the authors keep using completion time as reaction time for the whole time until the discussion which is the end of the manuscript!

Reply: Thank you for pointing it out. We have added the spelled-out and the short form in the abbreviation.

21. Reference:

Castro-Alonso, J. C., Ayres, P., & Paas, F. (2014). Dynamic Visualisations and Motor Skills. In W. Huang (Ed.), Handbook of Human Centric Visualization (pp. 551-580). New York, NY: Springer New York.

Wong, A, Marcus, N, Ayres, P, Smith, L, Cooper, G.A, Paas, G.W.C, & Sweller, J. (2009). Instructional animations can be superior to statics when learning human motor skills. Computers in Human Behavior, 25(2), 339–347. doi:10.1016/j.chb.2008.12.012

Wong, A., Leahy, W., Marcus, N., & Sweller, J. (2012). Cognitive load theory, the transient information effect and e-learning. Learning and Instruction, 22(6), 449-457. doi:10.1016/j.learninstruc.2012.05.004

 Reply: Thank you for the references and we have cited them in the manuscript.

---

## [Decision Letter · Decision Letter 1]

4 Sep 2020

PONE-D-20-08555R1

Origami folding: Memory load effects on acquisition of sequential skills

PLOS ONE

Dear Dr. Zhao,

Thank you for submitting your manuscript to PLOS ONE. After careful consideration, we feel that it has merit but does not fully meet PLOS ONE’s publication criteria as it currently stands. Therefore, we invite you to submit a revised version of the manuscript that addresses the points raised during the review process. 

Please address this point made by Reviewer 1: "The level of learners' expertise may be a confounding factor in the design", and all points made by Reviewer 2. In this second revision, I might not ask for another round of external peer reviewing to decide about this interesting manuscript. 

We look forward to receiving your revised manuscript.

Kind regards,

Juan Cristobal Castro-Alonso, Ph.D.

Academic Editor

PLOS ONE

Reviewers' comments:

Reviewer's Responses to Questions

**Comments to the Author**

1. If the authors have adequately addressed your comments raised in a previous round of review and you feel that this manuscript is now acceptable for publication, you may indicate that here to bypass the “Comments to the Author” section, enter your conflict of interest statement in the “Confidential to Editor” section, and submit your "Accept" recommendation.

Reviewer #1: (No Response)

Reviewer #2: All comments have been addressed

2. Is the manuscript technically sound, and do the data support the conclusions?

Reviewer #1: Partly

Reviewer #2: Partly

3. Has the statistical analysis been performed appropriately and rigorously? 

Reviewer #1: Yes

Reviewer #2: I Don't Know

4. Have the authors made all data underlying the findings in their manuscript fully available?

Reviewer #1: Yes

Reviewer #2: Yes

5. Is the manuscript presented in an intelligible fashion and written in standard English?

Reviewer #1: Yes

Reviewer #2: Yes

6. Review Comments to the Author

Reviewer #1: Thank for the revised version. I really appreciate the revisions, however, I still have concern about my first comment on the design issue. The level of learners' expertise may be a confounding factor in the design....

Would authors like to run another experiment with between-subject design? Although authors have put my concern as limitation, but the reported experiment may have design issues given no another experiment.

Reviewer #2: Thank you for the revision.

This version is made clearer that the authors are actually investigating in the type of cognitive resources rather than the cognitive load induced by learning origami. The previous version focused on the cognitive load which is a slightly different concept from resources. It is recommended to also change the keywords (and probably also the title) so not to confuse other new readers.

Moreover, another question arose – was the study going to test the cognitive resources that is relevant to learning *motor task through visualisation*, or to learning *procedural skills* as the author stated multiple times? If the aim is the first one, then the experimental setting makes sense. The secondary tasks were visuospatial-related that were relevant to the learning tasks (i.e. motor tasks) and the learning means (i.e. visualisations). Moreover, the author stated a multiple time in the article and also in the reply to the first author that the aim is to study the learning of procedural skills. Then in the experimental setting, the secondary task should be procedural- or sequential- related. That means the secondary tasks should be some tasks that might hinder participants from memorising the correct sequence. Otherwise the conclusion would not follow from the design.

This is also the reason why I put "I don't know" about the rigour of the statistical analysis, as the analysis method largely follows from the study aim and hypothesis.

Also, authors replied to the first reviewer that this is the first study – maybe it’s true when looking at learning origami under repetitive practices; but it’s definitely not the first study dealing with acquisition of sequential skill through looking at origami (see the reference that I recommended before).

In this revised version, there are still some issues with sentence structures. For example, line 52 “when participants concurrently solve[ing] a cognitive memory load task” What is a cognitive memory load task? Cognitive memory load is not an adjective. What type of tasks that you are referring to? Another example in line 55 “Yet an impact of load on performance does not necessarily imply an impact on learning (i.e., improvement of performance across trials with practice)” i.e. (id est) means “in other words, such as”. What does it mean by no impact on learning such as improvement of performance? Or as simple as missing a subject after adjustive in line 75 “As folding Origami can require cognitive as well as motor control,”…etc. With no intention to list out all the language issue, it is recommended to undergo another thorough proofread before re-submission.

7. PLOS authors have the option to publish the peer review history of their article (what does this mean?). If published, this will include your full peer review and any attached files.

Reviewer #1: No

Reviewer #2: No

---

## [Author Response · Author response to Decision Letter 1]

17 Sep 2020

Manuscript: PONE-D-20-08555R1

Title: Origami folding: Taxing resources necessary for the acquisition of sequential skills

Journal: PLOS ONE

Date: 16-Sep-2020

Response to Reviewer #1

1. Thank for the revised version. I really appreciate the revisions, however, I still have concern about my first comment on the design issue. The level of learners' expertise may be a confounding factor in the design....

Would authors like to run another experiment with between-subject design? Although authors have put my concern as limitation, but the reported experiment may have design issues given no another experiment.

Reply: Thank you for pointing out the potential benefits of a study involving experts. As we were to study the (early parts of) acquisition of Origami folding, we tested novices as they (due to lack of prior knowledge) should show the largest learning gains and they should be affected the most by taxing the cognitive resources potentially relevant for acquiring the sequential skill. By testing participants with expertise, the potential to find out which resources are relevant for acquiring the skill (by trying different secondary tasks) might have been more limited. Experts might rely on many (sub)processes that are automatized (i.e., no longer affected by taxing the resources by a secondary task). 

Given that we used a within-subjects design and we largely avoided that between subjects differences might confound our results. Nevertheless, combining a study on (early) practice in Origami folding with an expert-novice comparison seems a promising route for a future study. We have added the suggestion in the discussion.

Response to Reviewer #2

1. This version is made clearer that the authors are actually investigating in the type of cognitive resources rather than the cognitive load induced by learning origami. The previous version focused on the cognitive load which is a slightly different concept from resources. It is recommended to also change the keywords (and probably also the title) so not to confuse other new readers.

Reply: Thank you for pointing this out. “Memory load” can be misleading, as we used variants of memory loads to tax specific resources. We have changed the keyword of “memory load” into “cognitive resources”. The title has been changed from "Origami folding: Memory load effects on acquisition of sequential skills" to "Origami folding: Taxing resources necessary for the acquisition of sequential skills".

2. Moreover, another question arose – was the study going to test the cognitive resources that is relevant to learning *motor task through visualisation*, or to learning *procedural skills* as the author stated multiple times? If the aim is the first one, then the experimental setting makes sense. The secondary tasks were visuospatial-related that were relevant to the learning tasks (i.e. motor tasks) and the learning means (i.e. visualisations). Moreover, the author stated a multiple time in the article and also in the reply to the first author that the aim is to study the learning of procedural skills. Then in the experimental setting, the secondary task should be procedural- or sequential- related. That means the secondary tasks should be some tasks that might hinder participants from memorising the correct sequence. Otherwise the conclusion would not follow from the design. This is also the reason why I put "I don't know" about the rigour of the statistical analysis, as the analysis method largely follows from the study aim and hypothesis.

Reply: Thank you for the comment. We took care to make clear that we assume that visual representations are essential in the skill acquisition we are studying. We now avoid framing this as a procedural skill. We used a secondary task interfering with the visuospatial sketchpad. Given that we study the early part of skill acquisition (rather than automation of motor patterns), it seems reasonable that task performance and learning are mainly guided by visual representations. In line with the theory by Willingham (1998) on the connection of learning and motor control, participants might mainly rely on (and learn based on) visual representations early in practice (i.e. within the scope of our experiment). In work on sequence learning, it has been shown that participants (at least early in practice) learn about which target in space to affect in which order (sequence of positions rather than sequence of finger movements; cf. Willingham et al. 2000). When practice extends across many sessions, finger movements might become more relevant. According to this reasoning, a visual spatial secondary task should be especially potent to interfere with encoding the sequence or retrieving encoded sequence information.

4. Also, authors replied to the first reviewer that this is the first study – maybe it’s true when looking at learning origami under repetitive practices; but it’s definitely not the first study dealing with acquisition of sequential skill through looking at origami (see the reference that I recommended before).

Reply: Thank you very much for the comment. We have edited this and include these studies. 

5. In this revised version, there are still some issues with sentence structures. For example, line 52 “when participants concurrently solve[ing] a cognitive memory load task” What is a cognitive memory load task? Cognitive memory load is not an adjective. What type of tasks that you are referring to? Another example in line 55 “Yet an impact of load on performance does not necessarily imply an impact on learning (i.e., improvement of performance across trials with practice)” i.e. (id est) means “in other words, such as”. What does it mean by no impact on learning such as improvement of performance? Or as simple as missing a subject after adjustive in line 75 “As folding Origami can require cognitive as well as motor control,”…etc. With no intention to list out all the language issue, it is recommended to undergo another thorough proofread before re-submission.

Reply: Thank you very much for the comment. The listed issues have been corrected. Moreover, we have checked the whole manuscript for grammatical and other language issues as well as for redundancies.

---

## [Decision Letter · Decision Letter 2]

23 Sep 2020

Origami folding: Taxing resources necessary for the acquisition of sequential skills

PONE-D-20-08555R2

Dear Dr. Zhao,

We’re pleased to inform you that your manuscript has been judged scientifically suitable for publication and will be formally accepted for publication once it meets all outstanding technical requirements.

Kind regards,

Juan Cristobal Castro-Alonso, Ph.D.

Academic Editor

PLOS ONE

Additional Editor Comments (optional):

Reviewers' comments:

Reviewer's Responses to Questions

**Comments to the Author**

1. If the authors have adequately addressed your comments raised in a previous round of review and you feel that this manuscript is now acceptable for publication, you may indicate that here to bypass the “Comments to the Author” section, enter your conflict of interest statement in the “Confidential to Editor” section, and submit your "Accept" recommendation.

Reviewer #2: All comments have been addressed

2. Is the manuscript technically sound, and do the data support the conclusions?

Reviewer #2: Yes

3. Has the statistical analysis been performed appropriately and rigorously? 

Reviewer #2: Yes

4. Have the authors made all data underlying the findings in their manuscript fully available?

Reviewer #2: Yes

5. Is the manuscript presented in an intelligible fashion and written in standard English?

Reviewer #2: Yes

6. Review Comments to the Author

Reviewer #2: Thank you very much for the revision.

Now the entire experimental setting is seemingly more cohesive with the purpose of the study, the hypotheses and the conclusion.

I am happy to recommend for an acceptance.

7. PLOS authors have the option to publish the peer review history of their article (what does this mean?). If published, this will include your full peer review and any attached files.

Reviewer #2: No

---

## [Editor Report · Acceptance letter]

25 Sep 2020

PONE-D-20-08555R2 

Origami folding: Taxing resources necessary for the acquisition of sequential skills 

Dear Dr. Zhao:

I'm pleased to inform you that your manuscript has been deemed suitable for publication in PLOS ONE. Congratulations! Your manuscript is now with our production department. 

Kind regards, 

on behalf of

Dr. Juan Cristobal Castro-Alonso 

Academic Editor

PLOS ONE